# Dusty Plasma in Inhomogeneous Magnetic Fields in a Stratified Glow Discharge

**DOI:** 10.3390/molecules26133788

**Published:** 2021-06-22

**Authors:** Elena S. Dzlieva, Lev G. D’yachkov, Leontiy A. Novikov, Sergey I. Pavlov, Viktor Y. Karasev

**Affiliations:** 1Faculty of Physics, Saint-Petersburg State University, Universitetskaya nab. 7/9, Saint-Petersburg 199034, Russia; e.dzlieva@spbu.ru (E.S.D.); leontiynovikov@gmail.com (L.A.N.); s.i.pavlov@spbu.ru (S.I.P.); 2Joint Institute for High Temperatures, Russian Academy of Sciences, Izhorskaya st. 13, Bld. 2, Moscow 125412, Russia; dyachk@mail.ru

**Keywords:** complex plasma, dust particle rotation, glow discharge, magnetic field

## Abstract

We study the dynamics of dust particles in a stratified glow discharge in inhomogeneous magnetic fields. Dust structures are formed in standing striations, in which traps for dust particles arise. When a magnetic field is applied, these structures begin to rotate. The observations were carried out in striations near the end of the solenoid, where the region of an inhomogeneous magnetic field begins. With an increase in the magnetic field, the dusty structure can be deformed. The rotation of a dusty structure in an inhomogeneous magnetic field has been studied in detail; it has its own peculiarities in comparison with rotation in a uniform field. We have considered the mechanisms of such rotation and estimated its velocity.

## 1. Introduction

Dusty plasma is an example of an open system in which dissipation processes are extremely large [1,2,3,4,5]. Damping of translational, rotational and oscillatory motions of dust particles occurs in a short time and on small spatial scales [6,7]. For example, the figure of merit of a dusty plasma as an oscillatory system has a value of only a few units. Without intensive feeding of energy, any movement of dust particles is impossible. The inflow of energy and momentum occurs as a result of interaction with ions and electrons of the plasma, which is associated with a large charge of the dust particle–up to 10${}^{3}$–10${}^{5}$ elementary charges. Usually this charge is negative due to the higher electron mobility. A high charge increases the action of all types of external forces: conservative (for example, electrostatic), dissipative (ion drag), and gyroscopic (Lorentz forces acting on a dust particle either directly or indirectly through the light components of the plasma).

For many applications, it is necessary to make dust particles more controllable and mobile. In particular, for this purpose, active particles are created using special coatings on their surfaces [8,9,10,11,12] which intensively accumulate energy and momentum, transforming them into directional motion and drift. External fields are also used, for example, a magnetic field [13,14,15,16,17,18,19]. For the experimental study of bulk dusty structures, fields with longitudinal gradients can be used; then forces of different magnitudes will act on different horizontal sections of the structure. During the last 20 years, dusty plasma in a magnetic field has been of interest, first of all, from the point of view of dynamics, in particular, the establishment of rotation mechanisms under various conditions [14,20,21]. Methods of controlling dusty structures by means of a magnetic field, which are associated with both the control of the shape and size of dusty structures and the arrangement of particles in them, were considered [14,20,21,22,23,24,25,26,27,28,29,30,31]. Interesting, in particular, are the results of studying the proper rotation of dust particles [32,33,34,35], which showed that the magnetic properties of dusty plasmas arise due to a change in the flux of plasma particles onto their surface. Precessional motions are found, and it is shown that dusty plasma has paramagnetic properties.

At present, in the study of complex plasmas, more and more strong magnetic fields of up to several Tesla are used. Basically, such studies are carried out under the conditions of an RF discharge, in which dusty plasma is usually created in the form of monolayers perpendicular to the magnetic field [15,31,36]. In DC discharges, when strong magnetic fields are applied, instabilities of various types arise, which interferes with the study of complex plasma, since it can lead to the destruction of dust structures and even the extinction of the discharge. In recent experiments [17,37,38], for the first time, it was possible to create a dusty plasma under the conditions of a stratified glow DC discharge in a field up to 2.2 T. The study of bulk dust formations in a magnetic field opens up a number of new possibilities. For example, plasma flow inhomogeneities formed in the presence of longitudinal and radial magnetic field gradients can be used for more selective control of dusty structures.

The aim of this work is to study the dynamics of a dusty plasma structure in a stratified glow discharge placed in an inhomogeneous magnetic field, and its theoretical interpretation. For this, the magnetic field magnitude was limited to values at which there is still no significant effect on the dust trap (standing striation), but the effect of the inhomogeneous field on dust particles is already manifested. In the experiment, the rotation velocities of the dusty structure cross sections were measured at different magnetic field inductions, varying along the axis of the discharge. In the experiment, the rotation velocities of the dusty structure cross sections have been measured at different magnetic field inductions, varying along the axis of the discharge. Various dusty structure rotation mechanisms are considered, and a theoretical model taking into account the magnetic field inhomogeneity is proposed. The main difficulty in the theoretical description and obtained results interpretation was that different rotation mechanisms operate in opposite directions.

## 2. Experiment

### 2.1. Set Up

The experiment was carried out with dusty structures formed in standing striations of a glow discharge in an inhomogeneous magnetic field near the solenoid upper end. The dusty plasma structure was created in a vertically oriented discharge tube 80 cm long with an inner diameter of 2.8 cm. The cathode was at the bottom and the anode at the top. The tube was placed in the hole of the solenoid. The hole diameter was 5.2 cm, the solenoid height was 14 cm, and its outer diameter was 25 cm. An experimental setup fragment is shown schematically in Figure 1. Figure 2 shows the distribution of the longitudinal component $B_{z}$ of the magnetic induction on the tube axis depending on the vertical coordinate *h* measured from the upper end of the solenoid for one of its operating modes (current in its winding). For all other modes of operation, we have the same dependence $B_{z}\left(h\right)$, only the scale on the ordinate axis changes. The $B_{z}$ measurements were carried out for some operating modes using magnetometer. For all operating modes, we performed the calculations of the axial $B_{z}$ and radial $B_{r}$ components of the magnetic field by integrating the Biot-Savart law. The results of measurements and calculations of $B_{z}$ coincide within the measurement error. The magnetic field vector B on the axis of the discharge tube is directed upward $\left(B_{z}>0\right)$.

The stratified discharge was created in helium at pressures of 0.27, 0.5, and 1.0 Torr and a discharge current from 1.1 to 1.5 mA. The anode was located in the upper part of the discharge tube, and the cathode was in the lower one. A conical dielectric insert was placed in the discharge tube, which narrows the discharge and is necessary to create standing striations. The dusty structure was created by injecting melamine formaldehyde (MF) particles 1.1 μm in diameter into the discharge from a container with a mesh bottom located in the upper part of the discharge tube. A semiconductor laser module with a wavelength of 650 nm and a power of 40 mW was used to illuminate the dust particles. A cylindrical lens converted the light from the module into a horizontal flat beam (laser knife). Observation and video recording of dust particles was carried out through the upper end window of the tube using a video camera with a shooting frequency of 25 frames per second. A camera with a CCD matrix, which had a resolution of 570 TV lines with interlaced scanning and a sensitivity of the order of $10^{-4}$ lx, was used in the experiment. It was equipped with a high-aperture mirror lens with a focal length of 200 mm. The shooting was carried out through an interference filter, which removes the discharge glow. This system allows us to image the scattered light on small particles of 1 μm in size with good quality.

We studied the rotation of dust particles in striations V, VI, and VII, the approximate position of which is shown in Figure 1 (the striations are numbered above the dielectric insert). Most of the observations were made in striation VI, located slightly above the end of the solenoid. Figure 3 shows an image of a rotating section of a dusty structure in a magnetic field in this striation.

The main difference between this experiment and experiments in a uniform magnetic field [29,30] is that, with a change in the magnetic field, the position of the striation (dust trap) changes, and with it the position of the dust structure observed, and the observation conditions change accordingly. Therefore, the dependence of the rotation velocity of the dusty structure on the magnetic field can be determined at relatively small intervals of the field variation. But at some values of the discharge and the magnetic field parameters, by moving the illumination, it was possible to make observations at once in two or three striations (V, VI, VII), one of which (V) was inside the solenoid. The periscopic illumination system described in [17,37,38] was used to illuminate the structure in striation (V) under the upper end of the solenoid.

### 2.2. Experimental Results

In the experiment, at the given parameters, the selected horizontal cross section was illuminated, its position *h* above the solenoid end and the striation length were determined, and then the particle rotation angular velocity was measured. The angular velocity was defined in the standard way as the ratio of the dust particle rotation angle to the rotation time. Its error was mainly specified by the inaccuracy in determining the rotation center, which, as a result of various random uncontrolled processes, could slightly deviate from the discharge tube axis. The most extended and stable dust structures were formed at a pressure p=0.27 Torr. In this case, the striation length was 2.7 cm. The dust structure length was about 4 mm, it consists of horizontal layers with an average distance between them of about 0.3 mm, which does not change when a magnetic field is applied. One of the layers is shown in Figure 3. The dust structure is located in the lower part of the striation (its luminous head), where there are conditions for the stable equilibrium position of dust particles. In Figure 4 we show the rotation angular velocity of several cross sections of the dust structure in striation VI. The rotation velocity of the upper layers in the dusty structure is somewhat higher in absolute value than the lower ones, and each of them rotates like an almost rigid structure. The dust structure rotation angular velocity vector Ω${}_{d}$ is directed downward, i.e., opposite to the magnetic induction vector B, which determines the positive rotation direction.

With increasing pressure, the striation length and the dust structure length decreased. At p=0.5 Torr, the striation length was 2.5 cm, while at p=1.0 Torr it was only 1.7 cm. Under these conditions, at the same magnetic field distribution near the solenoid end, it was possible to determine the dust structure rotation velocities in two (at p=0.5 Torr) or three (at p=1 Torr) striations at once. The results of these measurements are shown in Figure 5.

We tried to detect the radial dependence of the dust particle rotation velocity in the horizontal section of the dust structure. Figure 6 shows the particle rotation velocity measurement results in striation VI at different distances from the rotation axis for the conditions corresponding to Figure 5. In all cases, some difference in rotation velocities turned out to be within the measurement error. As shown in the next section, theoretical estimates, at least in the central part of the discharge tube, also do not give a dependence of the rotation velocity on the radius.

## 3. Dust Structure Rotation Mechanisms and Theoretical Estimations

Let us consider the mechanisms that cause dust particles to rotate in an inhomogeneous magnetic field. These are the same mechanisms that operate in a homogeneous magnetic field, but their own peculiarities also appear. The rotation of dust particles in a stratified discharge occurs together with the gas and relative to the gas under the action of ion drag. At relatively small fields $B≲10^{2}$ G, the main rotation mechanism is ion drag. This mechanism is discussed in detail in our paper [38]. It causes dust particles to rotate counterclockwise if viewed in the direction of the magnetic field. In stratified discharges, a different rotation mechanism begins to dominate with increasing magnetic field. Under the action of eddy currents arising in the striations, the gas rotates in the opposite direction. In fields of the order of $10^{3}$ G, this mechanism becomes predominant and rotation inversion occurs [21,30]. In the magnetic fields used in this work, both rotation mechanisms can be significant.

Let us consider first the mechanism of ion drag. The ion velocity in the direction perpendicular to the magnetic field is determined by the equation (see e.g., [39])
(1)$$u_{i}=\frac{\frac{e}{\left|B\right|}\left[E\times B\right]+\frac{1}{n_{i}\left|B\right|}\left[B\times \nabla \left(n_{i}T_{i}\right)\right]}{m_{i}\omega_{iB}\left(1+\nu_{ia}^{2}/4\omega_{iB}^{2}\right)},$$
where $m_{i}$, $n_{i}$, $T_{i}$, $\omega_{iB}$ and $\nu_{ia}$ are the ion mass, number density, temperature, cyclotron frequency and frequency of collisions with atoms. The first term in (Equation 1) is due to the ion drift in the crossing electric **E** and magnetic **B** fields. The second term is associated with the gradient of the ion pressure (diamagnetic ion current).

In a homogeneous axial magnetic field $B_{z}$, the ion azimuthal velocity is
(2)$$u_{i}\left(r\right)=\frac{-eE_{r}\left(r\right)+T_{i}n_{i}^{-1}\left(r\right)dn_{i}/dr}{m_{i}\omega_{iB}\left(1+\nu_{ia}^{2}/4\omega_{iB}^{2}\right)},$$
where $E_{r}$ is the radial ambipolar electric field. In Equation (Equation 2) we take into account that the ion temperature is close to the atom temperature $T_{a}$ and practically constant. Using well-known equations for the ambipolar electric field [39,40] and approximating the ion radial density distribution $n_{i}\left(r\right)$ in the discharge tube by the Bessel profile, for the axial region, where the dust structure is located, we get [38]
(3)$$u_{i}\left(r\right)=-\frac{4\omega_{iB}T_{e}\nu_{ea}}{m_{i}\nu_{ea}\left(\nu_{ia}^{2}+4\omega_{iB}^{2}\right)+2m_{e}\nu_{ia}\left(\nu_{ea}^{2}+\omega_{eB}^{2}\right)}\frac{2.9r}{R^{2}},$$
where $m_{e}$, $T_{e}$, $\omega_{eB}$ and $\nu_{ea}$ are the electron mass, temperature, cyclotron frequency and frequency of collisions with atoms, *R* is the tube radius.

In an inhomogeneous magnetic field its lines diverge (or converge), and a radial component of the field $B_{r}$ appears. Then in Equation (Equation 2) we have two additional terms:(4)$$u_{i}\left(r\right)=\frac{-eE_{r}+eE_{z}B_{r}/B+T_{i}n_{i}^{-1}\left(r\right)\left[\partial n_{i}/\partial r-\left(B_{r}/B\right)\partial n_{i}/\partial z\right]}{m_{i}\omega_{iB}\left(1+\nu_{ia}^{2}/4\omega_{iB}^{2}\right)}.$$

Here we took into account that in the axial region, where the dust structure is located, $B_{z}\ll B_{r}$ and then $B_{z}≅B$.

The ion drag force applied to a dust particle of radius *a* and charge $eZ_{d}<0$ can be estimated as [41]
(5)$$F_{i}\left(z\right)=\frac{8}{3}\sqrt{2\pi T_{i}m_{i}}a^{2}n_{i}\left(1+\frac{1}{2}z_{d}\tau +\frac{1}{4}z_{d}^{2}\tau^{2}\Pi \right)\left(u_{i}-u_{d}\right),$$
where $\tau =T_{e}/T_{i}$ and $z_{d}=\left|Z_{d}\right|e^{2}/aT_{e}$ is the dimensionless charge of the dust particle, Π is the modified Coulomb logarithm integrated over the ion velocity distribution function [41,42], and $u_{d}$ is the dust particle velocity. The force $F_{i}$ is balanced by the neutral gas friction force
(6)$$F_{a}=\frac{8}{3}\sqrt{2\pi T_{a}m_{a}}a^{2}n_{a}\left(u_{a}-u_{d}\right),$$
where $m_{a}$, $n_{a}$ and $u_{a}$ are the neutral atom mass, number density and velocity. Taking into account that $z_{d}∼1$, Π∼1, $T_{i}\approx T_{a}$, $m_{i}=m_{a}$ and $u_{d}\ll u_{i}$, we obtain
(7)$$u_{d}\left(r\right)≅\frac{\tau^{2}n_{i}}{4n_{a}}z_{d}^{2}u_{i}\left(r\right)+u_{a}\left(r\right).$$

We substitute (Equation 4) in (Equation 7) and go to the angular velocity, dividing the result by the radius:(8)$$\begin{array}{c} \Omega_{d}=\frac{u_{d}\left(r\right)}{r}≅-\frac{2.9n_{i}\left(0\right)}{R^{2}n_{a}}\frac{T_{e}z_{d}^{2}\tau^{2}\nu_{ea}\omega_{iB}}{m_{i}\nu_{ea}\left(\nu_{ia}^{2}+4\omega_{iB}^{2}\right)+2m_{e}\nu_{ia}\left(\nu_{ea}^{2}+\omega_{eB}^{2}\right)}+\\ +\frac{z_{d}^{2}\tau^{2}\omega_{iB}dB_{r}/dr}{n_{a}m_{i}B\left(\nu_{ia}^{2}+4\omega_{iB}^{2}\right)}\left(n_{i}\left(0\right)eE_{z}-T_{i}\frac{\partial n_{i}}{\partial z}\right)+\frac{u_{a}}{r}. \end{array}$$

The first term in (Equation 8) is negative and the same as in a homogeneous magnetic field [38]. The second term arises in an inhomogeneous field; here we take into account that near the discharge axis, where the dusty structure is located, we can put $n_{i}\left(r\right)=n_{i}\left(0\right)$, $B_{r}\left(r\right)=rdB_{r}/dr$ and $dB_{r}/dr$ = const. The last term $u_{a}/r$ is associated with the rotation of dust particles together with a gas of neutral atoms. In our experimental conditions, the field $E_{z}$ is directed downward ($E_{z}<0$), and $\left|n_{i}\left(0\right)eE_{z}\right|\gg \left|T_{i}\partial n_{i}/\partial z\right|$; therefore, the second term in (Equation 8) is negative, since in the region above the solenoid, where the dusty structure is located, $dB_{r}/dr>0$. Thus, the inhomogeneity of the magnetic field increases the negative rotation velocity of dust particles relative to the neutral gas.

If the dust particle structure is below the solenoid, where the magnetic field lines converge and $B_{r}<0$, then the term associated with the inhomogeneity of the magnetic field would be positive.

Rotation of dust particles together with gas under the action of a magnetic field was observed in discharges of various types [18,23,26,43,44,45]. In stratified dc discharges the gas rotation occurs under the influence of the eddy currents, which arise in striations in the presence of axial $T_{e}$ gradient and radial $n_{e}$ gradient [46,47,48,49]
(9)$$b_{e}\nabla n_{e}\times \nabla T_{e}=-\nabla \times j,$$
where $b_{e}=e/m_{e}\nu_{ea}$ is the electron mobility, $\nu_{ea}=n_{a}\sigma_{ea}{\left(T_{e}/m_{e}\right)}^{1/2}$ is the frequency of electron-atom collisions and $\sigma_{ea}$ is the electron-atom scattering cross section. The Ampere force in the longitudinal magnetic field, acting on the eddy current radial component, causes the gas to rotate. It is equilibrated by the gas viscosity [47,48]
(10)$$\eta \frac{\partial^{2}u_{\varphi }}{\partial r^{2}}+j\times B=0,$$
where η is the gas-dynamic viscosity, for helium $\eta =2\times 10^{4}$ g/(cm s).

We have presented a simple analytical model for eddy currents and their influence on the gas rotation in axial magnetic field $B_{z}$ [17]. It takes into account the modulation of electric field, electron and ion densities and electron temperature in a striation:(11)$$E\left(z\right)=\gamma E_{0}\left[1+k_{E}\exp\left(\beta \cos\left(2\pi z/L\right)\right)\right],$$
(12)$$n_{e}\left(z\right)=n_{e0}\left[1-k_{n}\cos\left(2\pi z/L\right)\right],$$
(13)$$T_{e}\left(z\right)=T_{e0}\left[1+k_{T}\cos\left(2\pi z/L\right)\right],$$
where *L* is the striation length. The parameters β, $k_{E}$ and γ in Equation (Equation 11) were chosen for good correlation with the field profile obtained under the similar experimental conditions [9]: β=3, $k_{E}=0.5$ and γ=0.29. The parameters $k_{n}$ and $k_{T}$ in Equations (Equation 11) and (Equation 12), according to the results of numerical calculation [50], should take values close to 0.5. In our estimates, we take the values $k_{n}=k_{T}=0.4$ used in [17]. A stable position of the dust structure is possible in the head of the striation, where the eddy current is directed towards the discharge axis. Here the rotation of the gas, and with it the dust particles, occurs in a positive direction. In contrast to [17], where the dusty structure was located inside the solenoid, in this experiment we can visually determine the striation length *L* and the dust structure position z/L in it. Using the simple eddy current model proposed in [17], we represent the last term in (Equation 8) in the following form
(14)$$\Omega_{a}=u_{a}/r=-0.0343R^{4}CB_{z}/\eta ,$$
where
(15)$$C\left(z\right)=\frac{\alpha^{2}b_{e}}{8R^{2}}n_{e0}T_{e0}k_{T}{\left(\frac{2\pi }{L}\right)}^{2}\left[-\cos\left(2\pi \frac{z}{L}\right)+k_{n}\cos\left(4\pi \frac{z}{L}\right)\right],$$
and α=2.405 is the first root of the Bessel function $J_{0}\left(x\right)$. The average electron density near the discharge axis can be estimated by the equation [17]
(16)$$n_{e0}=\frac{\alpha n_{a}\sigma_{ea}I{\left(m_{e}T_{e0}\right)}^{1/2}}{2\pi J_{1}\left(\alpha \right)R^{2}e^{2}E_{0}}.$$

In the head of the striation at z/L∼0.1 we have C<0 and then $\Omega_{a}>0$.

Equation (Equation 14) corresponds to the contribution of the axial component of the magnetic field $B_{z}$ to the gas rotation, and with it the rotation of dust particles. However, if there is also a radial component $B_{r}$, the vector product j×B in Equation (Equation 10) has a component $-j_{z}B_{r}$, where $j_{z}$ is the discharge current density. Accordingly, we have an additional component of the rotation velocity $u_{\varphi r}$, which is determined by the equation
(17)$$\eta \frac{\partial^{2}u_{\varphi r}}{\partial r^{2}}-j_{z}B_{r}=0.$$

For the Bessel profile of the charged particle density we have $j_{z}\left(r\right)=j_{z}\left(0\right)J_{0}\left(\alpha r/R\right)$, and then the discharge current can be presented as
(18)$$I=-2\pi \int_{0}^{R}j_{z}\left(r\right)rdr=-2\pi j_{z}\left(0\right)R^{2}J_{1}\left(\alpha \right)/\alpha ,$$
correspondingly
(19)$$j_{z}\left(r\right)=-\frac{I\alpha J_{0}\left(\alpha r/R\right)}{2\pi J_{1}\left(\alpha \right)R^{2}},$$

The minus sign in Equations (Equation 18) and (Equation 19) means that the vector j is directed downward (opposite to vector B), and by *I* we mean the absolute value of the discharge current. Substituting Equation (Equation 19) into Equation (Equation 17), after integration we obtain
(20)$$u_{\varphi r}\left(r\right)=-\frac{IRdB_{r}/dr}{2\pi \alpha^{2}J_{1}\left(\alpha \right)\eta }\int_{0}^{\alpha r/R}J_{1}\left(x\right)xdx+c_{1}r+c_{2}.$$

From the boundary conditions $u_{\varphi r}\left(0\right)=u_{\varphi r}\left(R\right)=0$ we get the integration constants: $c_{2}=0$ and
(21)$$c_{1}=\frac{IdB_{r}/dr}{2\pi \alpha^{2}J_{1}\left(\alpha \right)\eta }\int_{0}^{\alpha }J_{1}\left(x\right)xdx=0.078\frac{I}{\eta }\frac{dB_{r}}{dr}.$$

As a result, an additional term appears in Equation (Equation 14):(22)$$\Omega_{a}=-0.0343R^{4}C\left(z\right)B_{z}/\eta +\frac{I}{\eta }\frac{dB_{r}}{dr}\left[0.078-0.053\frac{R}{r}\int_{0}^{\alpha r/R}J_{1}\left(x\right)xdx\right],$$

Thus, the angular velocity of dust particle rotation has four components, two of them arise due to the presence of a longitudinal magnetic field, the other two–when a radial component appears, i.e., in an inhomogeneous magnetic field. The sign of the last two depends on the direction $B_{r}$. Near the discharge tube axis, where the dusty structure is located, the radial component of the magnetic field $B_{r}$ is small in comparison with $B_{z}$. Therefore, the contribution of the corresponding terms to the total rotation velocity is not large. If $B_{r}>0$ (at the output of the solenoid), they have the same signs as the corresponding terms due to the longitudinal magnetic field $B_{z}$; if $B_{r}<0$, their signs are opposite.

In our experimental conditions $T_{i}\approx 300$ K, $T_{e0}/T_{i}\approx 10^{2}$. For helium we take the gas-kinetic atomic scattering cross section $\sigma_{ea}\approx 2\times 10^{-15}$ cm${}^{2}$ and resonant charge exchange cross section of ion He${}^{+}$$\sigma_{ia}\approx 4\times 10^{-15}$ cm${}^{2}$ [51]. For brevity, let us denote the first and second terms in Equation (Equation 8) due to the ion drag in the axial and radial magnetic fields, respectively, by $\Omega_{1}$ and $\Omega_{2}$. Similarly, in Equation (Equation 22), which determines the gas rotation velocity in the head of the striation, we denote the first and second terms by $\Omega_{3}$ and $\Omega_{4}$, respectively. As noted above $\Omega_{1}<0$ and $\Omega_{3}>0$. The measurements of the rotation velocity were carried out in the region where $B_{r}>0$, therefore $\Omega_{2}<0$ and $\Omega_{4}>0$.

The main problem when comparing the theoretical estimates of the rotation velocity with experimental data under the considered conditions is that the rotation of the gas and the rotation of dust particles under the action of ion drag occur in opposite directions with velocities of the same order of magnitude. Therefore, the relative error of the result can be large and exceed the rotation velocity. In addition, some parameters required for estimates are known approximately. But we can choose the values of these parameters, without going beyond the permissible limits, in order to obtain reasonable agreement with the experiment. In our estimates, we take the dimensionless charge of a dust particle $z_{d}$ as such a parameter. Under our experimental conditions, for micron-sized dust particles in helium, $z_{d}$ is about 2 or some less [41,52]. It should be noted that only $\Omega_{1}$ and $\Omega_{2}$ depend on $z_{d}$.

The selected $z_{d}$ values and the results of estimating the angular velocity $\Omega_{d}$ and all its components $\Omega_{i}$ (*i* = 1–4) for the experimental conditions are shown in Table 1. We see that $\Omega_{1}+\Omega_{2}$ and $\Omega_{3}+\Omega_{4}$ are close in absolute value and opposite in sign, their absolute values can exceed the experimental result by an order of magnitude. To show the strong dependence of $\Omega_{d}$ on the choice of $z_{d}$, we compare the $\Omega_{1}$, $\Omega_{2}$ and $\Omega_{d}$ values at $z_{d}=2.15$ for the conditions at p=0.27 Torr, I=1.4 mA and h=7 mm with those presented in the first row in Table 1 for $z_{d}=2.25$. In this case $\Omega_{1}=-4.84$, $\Omega_{2}=-2.76$ and $\Omega_{d}=-0.8$. Thus, a 0.1 (<5%) change in $z_{d}$ leads to a slight change in $\Omega_{1}$ and $\Omega_{2}$ (∝10%), but radically changes the total rotation velocity $\Omega_{d}$. Another important point should be noted. In our estimations, we use the eddy current model proposed by us in [17]. However, alternative models are possible that slightly change the $\Omega_{3}$ calculation, and the $z_{d}$ value should be changed accordingly. Dust particles are located at the head of the striation, where there is a region of stable equilibrium for them (dE/dz<0). Here, with an increase in the *z* coordinate and a decrease in the field *E*, the particle charge should also decrease. When we observe the rotation of dust particles at different heights in the same striation (p=0.27 Torr), our estimate of their charges corresponds to such a dependence of the charge on the height.

## 4. Conclusions

The dynamics of dust structures in the plasma of a stratified glow discharge under the action of an inhomogeneous magnetic field with a longitudinal gradient of $dB_{z}/dz≲10^{2}$ G/cm has been studied experimentally and theoretically. An inhomogeneous field arises above the end face of the solenoid, where the magnetic field lines diverge, and a radial field component appears. Dusty plasma is formed in standing striations in this area. Under the influence of a magnetic field, dust structures start to rotate. Their angular rotation velocity was measured at different distributions of the longitudinal and radial magnetic field components. The angular velocity vector direction is opposite to the magnetic field vector at all values of the magnetic field up to ∼500 G.

The dust structure rotation mechanisms in a stratified glow discharge under action of an inhomogeneous magnetic field are considered in detail. These are drag by ions drifting in crossed magnetic and electric fields, as well as drag by a neutral gas, whose rotation in a magnetic field is associated with eddy currents in striations. The same mechanisms operate in uniform magnetic fields, but in inhomogeneous fields, each of them makes an additional contribution associated with the presence of a radial magnetic field. These mechanisms operate in opposite directions. As a result, the total rotation velocity is the result of the addition of four components with different signs, which in the conditions under consideration are close in absolute value. It should be noted that good quantitative agreement between theoretical estimates and experimental data under such conditions was obtained with an appropriate choice of one of the parameters of the theory. However, this choice was carried out within the limits admissible for this parameter, and under conditions when many other parameters of the problem are known approximately. We want to emphasize that the main result of the theoretical part of our investigation is the identification of the dust structure rotation mechanisms in the conditions under consideration, rather than the quantitative agreement of theoretical estimations with experimental data.

## Figures and Tables

**Figure 1 molecules-26-03788-f001:**
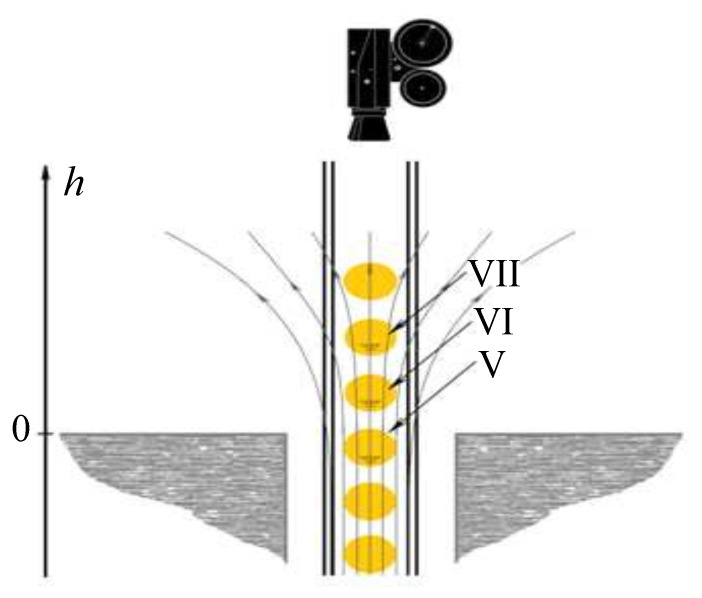
Sketch of the upper part of the experimental setup. A stratified glow discharge is ignited in the tube inside the solenoid. We observe dust structure rotation in the striations V, VI and VII near the upper end of the solenoid. They are shown at the bottom of these striations.

**Figure 2 molecules-26-03788-f002:**
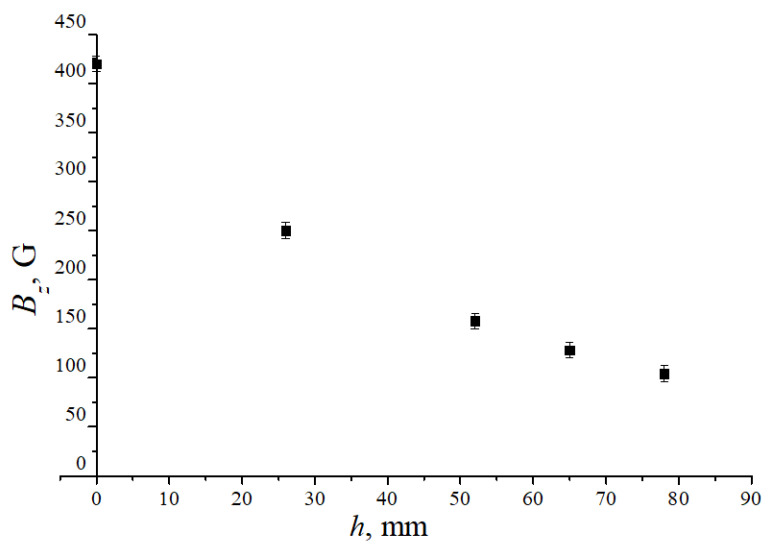
The measured distribution of the longitudinal component of the magnetic induction on the solenoid axis depending on the distance *h* from its upper end.

**Figure 3 molecules-26-03788-f003:**
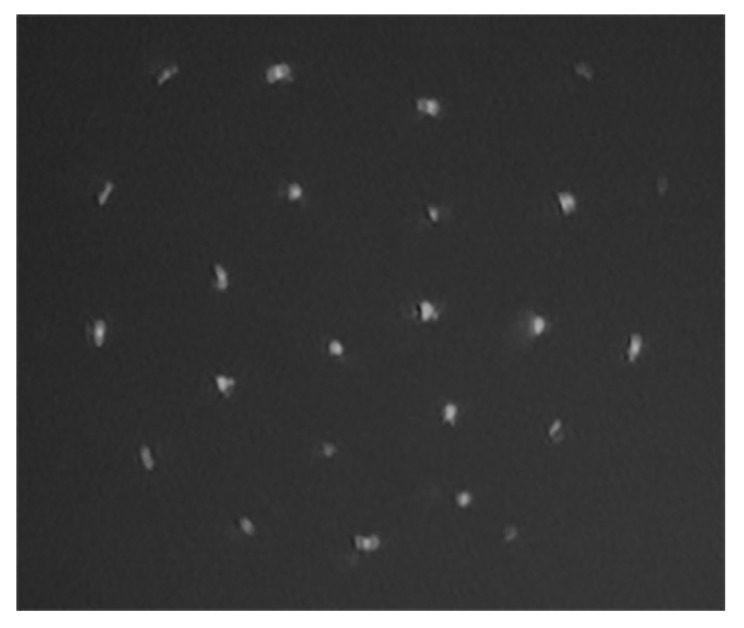
Horizontal section of a rotating dust structure in striation VI at h=8.5 mm under conditions: p=0.27 Torr, discharge current I=1.5 mA, B=390 G. The horizontal size of the frame is 2.4 mm.

**Figure 4 molecules-26-03788-f004:**
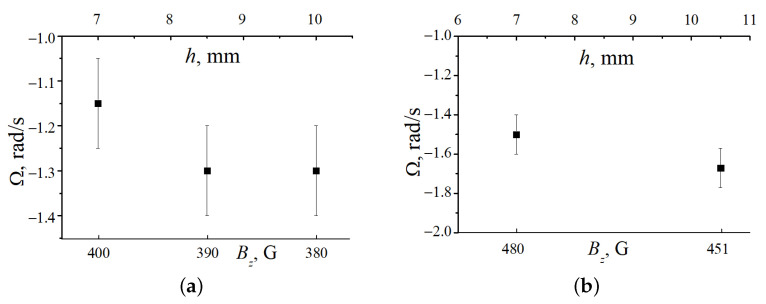
Dependences of the angular velocity of the dust structure rotation in striation VI on the vertical coordinate *h* at p=0.27 Torr. (**a**) I=1.5 mA, on the interval of 7–10 mm, the magnetic field varies from 400 to 380 G. (**b**) I=1.4 mA, in the interval of 7–10.5 mm, the magnetic field varies from 480 to 451 G.

**Figure 5 molecules-26-03788-f005:**
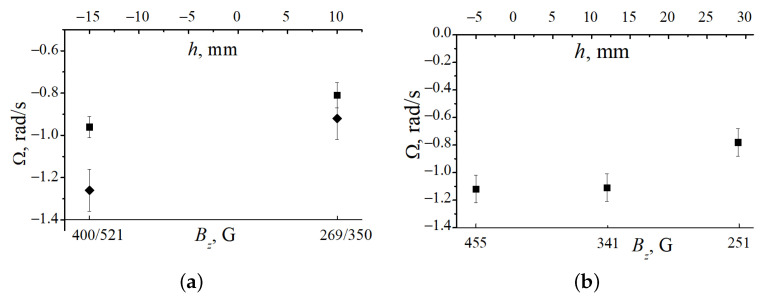
Angular velocities of dust structures in two (**a**) or three (**b**) striations, depending on their position *h* near the solenoid upper end. (**a**) p=0.5 Torr; squares: I=1.2 mA, structures are located in striation V (h=−15 mm, B=400 G) and striation VI (h=10 mm, B=269 G); rhombs: I=1.3 mA, structures are located in striation VI (h=−15 mm, B=521 G) and striation VII (h=10 mm, B=350 G). (**b**) p=1 Torr, I=1.2 mA, structures are located in striation V (h=−5 mm, B=455 G), striation VI (h=12 mm, B=341 G) and striation VII (h=29 mm, B=251 G).

**Figure 6 molecules-26-03788-f006:**
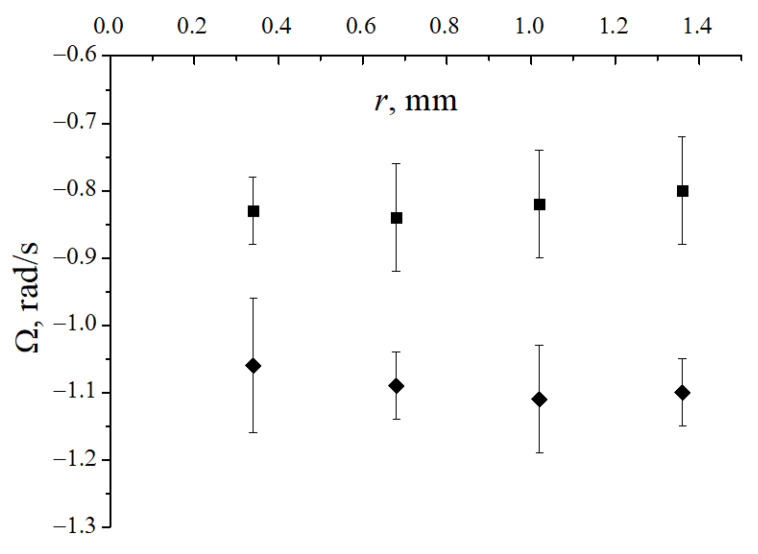
Dependence of the angular velocity of dust particles on the radius *r* at p=0.5 Torr, h=10 mm. Squares: striation VI, I=1.2 mA, B=269 G; rhombs: striation VII, I=1.3 mA, B=350 G.

**Table 1 molecules-26-03788-t001:** Angular velocity and its components for a dusty structure in magnetic fields. Comparison of estimations with experimental data.

*h*,	$\Omega_{\exp}$	$B_{z}$,	${\mathrm{dB}}_{r}/\mathrm{dr}$,	z/L	$z_{d}$	$\Omega_{1}$	$\Omega_{2}$	$\Omega_{3}$	$\Omega_{4}$	$\Omega_{d}$
mm		G	G/cm							
p=0.27 Torr, L=2.7 cm, I=1.4 mA										
7	−1.5±0.1	480	41.9	0.17	2.25	−5.31	−3.03	4.53	2.27	−1.53
10.5	−1.67±0.1	451	40.0	0.3	1.68	−2.82	−1.14	0.10	2.17	−1.69
p=0.27 Torr, L=2.7 cm, I=1.5 mA										
7	−1.15±0.1	400	34.9	0.17	2.15	−4.56	−2.30	3.71	2.03	−1.13
8.5	−1.3±0.1	390	34.2	0.22	1.99	−4.61	−1.77	3.06	1.98	−1.33
10	−1.3±0.1	380	33.6	0.28	1.73	−3.22	−1.11	1.06	1.95	−1.32
p=0.5 Torr, L=2.5 cm, I=1.2 mA										
−15	−0.96±0.06	400	26.4	0.22	1.85	−6.06	−0.83	4.72	1.23	−0.94
10	−0.81±0.06	269	23.8	0.22	1.70	−4.43	−0.65	3.18	1.1	−0.80
p=0.5 Torr, L=2.5 cm, I=1.3 mA										
−15	−1.28±0.1	521	34.4	0.22	2.16	−7.59	−1.40	6.00	1.73	−1.26
10	−1.09±0.1	350	31.8	0.22	1.92	−5.66	−1.06	4.03	1.6	−1.09
p=1 Torr, L=1.7 cm, I=1.2 mA										
−5	−1.12±0.1	455	35.6	0.24	2.22	−19.24	−1.4	17.97	1.65	−1.03
12	−1.11±0.1	341	30.4	0.24	2.09	−14.88	−1.09	13.48	1.41	−1.09
29	−0.78±0.1	251	22.7	0.24	1.99	−10.98	−0.75	9.91	1.05	−0.77

## Data Availability

The data that support the findings of this study are available from the corresponding author, upon reasonable request.

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
