# Peer review of "Dusty Plasma in Inhomogeneous Magnetic Fields in a Stratified Glow Discharge"

_molecules, 2021, doi:10.3390/molecules26133788_

Round 1
Reviewer 1 Report
The authors present a simple analytical model which can explain the physical processes driving the rotation of dust structures in an inhomogeneous magnetic field. The article is generally well-written with sound physical principles. As I read through the article, I did have a few questions, which I have tried to group together by topic. Numbers refer to line numbers in the draft manuscript. With minor clarifications, I recommend this manuscript for publication.
Observations within different striations / Experimental setup
98-101 (and 119-122). It is mentioned that observations in multiple striations can be made by “moving and rotating” the illumination. Can the laser sheet be moved vertically? How is it rotated? I am unclear how you can image a plane of the dust structure below the upper edge of the solenoid. In this case, is the imaged dust plane horizontal or at an angle? (If tilted, is this accounted for in the calculation of angular velocity?) Perhaps this could be indicated in Figure 1.
- Is the dust structure length the vertical extent of the dust cloud or the horizontal extent of the dust cloud? I assume that since you mention the “striation length” that this is the vertical extent of the dust cloud, yet your camera is imaging a horizontal plane. How is the vertical extent measured/recorded?
- Is the head of the striation the upper end (away from the solenoid) or the lower end? Perhaps this is indicated in Figure 1, where it appears that the dust is at the lower edge. Mention this in the figure caption as well as the text.
Taking the “length” to be the vertical extent of the dust structure, are measurements made for various planes within the dust structure? From the data shown in Figure 6, it appears that the dust in a horizontal plane rotates as a rigid structure, and I was wondering if this is true throughout the entire vertical extent of the dust structure. In ref [15] it is noted that the rotation is measured at the widest point of the dust structure. Is that true in this case as well? What is the vertical spacing between planes in the dust structure?
Magnetic Field
Figure 4. Can you add a second horizontal axis to the figures to show the variation of B?
Question: Figures 4a and 4b cover the same vertical extent above the upper edge of the solenoid. Why is the magnetic field different for the two pressure cases? Is this due to coupling with the plasma? How is the magnetic field calculated and /or measured?
180-182 – Here it is finally mentioned how the B field is calculated. How is it measured? Where is it measured? I take it the measurements were used to validate the calculations? It might help to give this information earlier in the manuscript.
Dust charge and Electric field in striations
214-215. It is mentioned that rotation of dust is observed within the same striation. Can you show this data? How do you estimate the charge with height?
Eddy height z/L – is this measured from the bottom of a striation or from the center of a striation?
Table 1. If you could assume that dE/dz is similar for all of the striations, is the change in z_d with z/L (as the z_d are selected to match the experimental rotational velocity) consistent with the model shown for the electric field in Eq. 11? Here I am assuming that qE = mg, but if ion drag in the vertical direction is significant, this is not true. I found an expression for the force balance and the calculation of z_d in your reference [15]. I think you should mention that reference in line 198 and the paragraph below discussing the change in charge with z/L, because otherwise the decrease in charge with the decrease in E(z) is counterintuitive. (This may be exacerbated by the fact that I am interpreting the direction of increasing z/L incorrectly.)
Minor comments:
Two sentences on lines 53-55 and 58-60 are repeated.
Eq. 9. Is “curlj” meant to be ∇ x j ?
- Typo – helium is written as “gelium”.
- “we present” the values for the components of the rotation velocity – this is confusing – I expected to see these values in the Table, Instead try “compare” (or “contrast”) the values.
Reviewer 2 Report
The authors present a combined experimental and theoretical study on the collective rotation properties of finite dust particle clouds in stratified DC discharges in inhomogeneous magnetic fields. The idea is interesting, as the rotation of dust clouds is a common phenomenon frequently observed in laboratory dusty plasmas. The fact that this rotation can be influenced by external magnets is also known, without deeper understanding.
The originality of the authors is the combination of experimental observation and a phenomenological model. The work is interesting and deserves publication, however, a few corrections and additions would improve its soundness.
These are:
The use of phrases like "For the first time..." emphasizing the originality of the work should be avoided as this kind of statement is not scientific and can not be proven.
In the experiments, dust particles with a diameter of 1.1 um are used. This is very small. Such small particles, illuminated with a 40 mW red laser beam that is expanded into a sheath, are practically invisible. A quick estimate gives something like 10-100 photons per pixel per image on the detector. This is below the detection limit of commercial digital cameras at room temperature. Also in fig. 3 the image of the particles shows a structure typical for dimer and multiple particles that are stuck together. This means that estimates for mass and charge are highly uncertain. Luckily, however, in the frame of the capacitor model linking diameter and charge, the model parameter z_d will not be influenced by this circumstance. Of course, the capacitor model is not exact, nor are the other pieces of the model constructed by the authors.
I suggest including a discussion of this point in the manuscript and emphasize that this model provides qualitative insight into the underlying processes, especially the competition of magnetic and convective effects, however, the model contains a highly uncertain parameter (z_d) and therefore has no quantitative predictive power.
Reviewer 3 Report
In this manuscript the results on particles dynamics in a stratified glow discharge in inhomogeneous magnetic field are reported. Dust structures are formed in standing striations and rotate when a magnetic field is applied. The rotation velocity is measured experimentally under different conditions and a theoretical model is proposed. Theory is compared with experiment.
The topic of this investigation is interesting and timely. The possibility to manipulate small particles in a plasma environment can potentially be relevant to many applications. Also, understanding basic plasma-particle interactions remains an important fundamental topic across disciplines. I am positively inclined towards recommending publication of this manuscript in Molecules.
However, in my opinion several improvements can be made. These are listed below:
- Introduction: In the context of active particles (lines 24-25) in plasma environment two recent papers by Nosenko et al. Phys. Rev. Research (2020) and Vasiliev et al. Molecules 25, 3375 (2020) should be apparently mentioned.
- Introduction: Last paragraph: Two sentences are duplicated in this paragraph.
- Figure 4 caption: “thbe” -> “the”
- (5): It would be helpful for the reader to provide references to the paper where this expression was originally derived. In particularly, this concerns the definition of the modified Coulomb logarithm.
- Line 198: The authors assume z_d is about 2. At the neutral gas pressures used in experiment I would expect that collision enhanced ion collection mechanism operates and z_d is about or less than unity (as for example observed in experiments with PK-4). I would suggest to discuss this point in more detail.
- Table I: Table I does not fit into the page vertically; Why not rotate it by 90 degrees?
- Conclusions on Lines 228-229: “entrainment” -> “drag”
